# Dysregulated Gut Homeostasis Observed Prior to the Accumulation of the Brain Amyloid-β in Tg2576 Mice

**DOI:** 10.3390/ijms21051711

**Published:** 2020-03-03

**Authors:** Pedram Honarpisheh, Caroline R. Reynolds, Maria P. Blasco Conesa, Jose F. Moruno Manchon, Nagireddy Putluri, Meenakshi B. Bhattacharjee, Akihiko Urayama, Louise D. McCullough, Bhanu P. Ganesh

**Affiliations:** 1Department of Neurology, University of Texas McGovern Medical School, Houston, TX 77030, USA; Pedram.Honarpisheh@uth.tmc.edu (P.H.); Caroline.R.Reynolds@uth.tmc.edu (C.R.R.); Maria.P.BlascoConesa@uth.tmc.edu (M.P.B.C.); Jose.Felix.MorunoManchon@uth.tmc.edu (J.F.M.M.); Akihiko.Urayama@uth.tmc.edu (A.U.); Louise.D.McCullough@uth.tmc.edu (L.D.M.); 2Department of Molecular and Cell Biology, Baylor College of Medicine, Houston, TX 77030, USA; Putluri@bcm.edu; 3Department of Pathology, University of Texas McGovern Medical School, Houston, TX 77030, USA; Meenakshi.B.Bhattacharjee@uth.tmc.edu

**Keywords:** gut-brain axis, neurodegenerative diseases, amyloid, Alzheimer’s disease, Tg2576, inflammatory cytokines, B12, tight junction proteins, gut integrity

## Abstract

Amyloid plaques in Alzheimer’s disease (AD) are associated with inflammation. Recent studies demonstrated the involvement of the gut in cerebral amyloid-beta (Aβ) pathogenesis; however, the mechanisms are still not well understood. We hypothesize that the gut bears the Aβ burden prior to brain, highlighting gut–brain axis (GBA) interaction in neurodegenerative disorders. We used pre-symptomatic (6-months) and symptomatic (15-months) Tg2576 mouse model of AD compared to their age-matched littermate WT control. We identified that dysfunction of intestinal epithelial barrier (IEB), dysregulation of absorption, and vascular Aβ deposition in the IEB occur before cerebral Aβ aggregation is detectible. These changes in the GBA were associated with elevated inflammatory plasma cytokines including IL-9, VEGF and IP-10. In association with reduced cerebral myelin tight junction proteins, we identified reduced levels of systemic vitamin B12 and decrease cubilin, an intestinal B12 transporter, after the development of cerebral Aβ pathology. Lastly, we report Aβ deposition in the intestinal autopsy from AD patients with confirmed cerebral Aβ pathology that is not present in intestine from non-AD controls. Our data provide evidence that gut dysfunction occurs in AD and may contribute to its etiology. Future therapeutic strategies to reverse AD pathology may involve the early manipulation of gut physiology and its microbiota.

## 1. Introduction

Cerebral amyloidopathy is a hallmark of neurodegenerative diseases (NDDs) including Alzheimer’s disease (AD) [1]. As a leading hypothesis in the pathogenesis of AD, the amyloid cascade hypothesis holds that amyloid-β (Aβ) aggregation in the central nervous system (CNS) is a primary etiology in AD and it is followed by neuroinflammation and neurotoxicity. In keeping with this hypothesis, evidence for the presence of a dysregulated neuroinflammatory milieu in AD was described nearly 20 years ago [2] and has been since documented in most NDDs [3,4]. However, new clinical and pre-clinical data have implicated inflammation and peripherally driven interactions as not only responses to, but also drivers of, AD pathogenesis [5]. However, whether peripheral inflammation is a causal factor or simply a secondary response to the development of cerebral amyloidopathy is unknown [6]. The dysregulation of gut homeostasis has been linked to various developmental [7,8] and non-developmental pathologies [9,10,11,12]. Intestinal epithelial barrier (IEB) dysfunction can induce changes in the host immune system, leading to significant reshaping of the gut microbiota to promote pathology, known as dysbiosis [13]. Conversely, IEB dysfunction and dysbiotic microbiota can act independently or synergistically to influence the immune response to gastrointestinal and extra-gastrointestinal pathologies [14,15,16,17].

The regulatory role of the gut–brain axis in NDDs is increasingly being recognized [18,19]. In 2011, using microbiota transplants, Bercik et al. showed that strain-specific behavioral traits in mice can be transmitted along with the microbiota [20] and Heijtz et al. showed that germ-free mice exhibit a less-anxious behavioral phenotype than conventionally raised mice [7]. In AD, studies using transgenic animal models demonstrated that cerebral Aβ pathology and neuroinflammation were significantly reduced in germ-free AD mice, when compared to conventionally raised AD mice [21,22]. Similarly, treatment with antibiotics reduced cerebral Aβ pathology and this effect was partially reversed by fecal microbiota transplant (FMT) from non-antibiotic-treated transgenic donors [23]. These results strongly suggest a link between gut and cerebral proteinopathies, including Aβ pathology. Several possible mechanisms have been suggested including production and propagation of microbial amyloid [24], systemic inflammatory response to microbial antigens [25,26], immune signaling cascades regulated by microbial metabolites [27,28], and modulation of the circulatory neurotransmitters or inflammatory cytokines [19,20,29,30].

In the present study, we hypothesized that pathogenesis of cerebral Aβ has a caudo-rostral gradient of dysfunction that begins as a dysregulation of IEB and peripheral inflammation. This peripheral disturbance might then spread centrally, leading to cerebral Aβ pathology. We used the Tg2576 transgenic mouse model of AD that overexpresses human APP695 containing the double mutation of K670N/M671L, under the control of hamster prion protein (PrP) [31]. Tg2576 mice develop cerebral Aβ aggregates, white matter injury, and cognitive impairment beginning around 9 months of age [32]. We examined these Tg2576 mice at young (Yg) 6 months (i.e., pre-symptomatic) and at aged (Ag) 15 months (i.e., symptomatic) timepoints. We demonstrate a significant impairment in IEB function, bacterial breach of the IEB, and disturbance in the plasma cytokine profile occur prior to the onset of cerebral Aβ depositions in Tg2576 mice. We identified significantly reduced cubilin [33] gene expression levels, a vitamin B12 transporter, in the intestinal epithelium that occurred prior to white matter injury in the CNS, in pre-symptomatic Yg-Tg2576 mice. In accordance with this decrease in cubilin, we found significantly reduced B12 levels in the blood plasma after Aβ deposition was present in symptomatic Tg2576 mice. Lastly, using autopsy samples of the brain and gut tissues from patients with AD diagnosis, we found that Aβ deposition is present, not only in the brain but also in the intestinal epithelium. Together, our results suggest that dysfunction of IEB and intestinal nutrient absorptions occur before development of cerebral pathology in Tg2576 mice, and potentially in human AD.

## 2. Results

### 2.1. Cerebral Aβ Plaques are Undetectable in Pre-Symptomatic Mice but Present in Symptomatic Tg2576 Mice

To confirm the presence of cerebral Aβ pathology and the timeline of plaque development, pre-symptomatic 6-month-old Tg2576 (Yg-Tg) and symptomatic 15-month-old Tg2576 (Ag-Tg) mice were examined by thioflavin S staining (Figure 1a). Consistent with previous reports on this mouse model [31,34,35,36], no parenchymal thioflavin S-positive plaques were seen in the brain at 6 months, whereas a significant plaque burden was detected at 15 months in the subiculum and hippocampus of Tg2576 mice, when compared to age-matched WT littermate controls (Figure 1b).

### 2.2. IEB Dysfunction Occurs before Development of Cerebral Aβ Pathology in Tg2576 Mice

Mucus is the primary protective barrier separating luminal antigens and the intestinal epithelium [37,38]. Mucus is secreted by goblet epithelial cells and is highly glycosylated [39]. Inflammation in the intestine depletes healthy fucosylated mucus [40]. We evaluated the mucus layer maturity by lectin staining and mucus fucosylation by *Ulex europaeus* agglutinin staining. Mucus fucosylation was significantly attenuated in the large intestine of the pre-symptomatic Tg2576 mice (Yg-Tg), compared to age-matched WT littermates (Yg-WT) (*p < 0.05*, Figure 2a,b). However, this reduction in intestinal mucus fucosylation did not persist in the symptomatic Tg2576 mice (Ag-Tg), compared to age-matched WT littermates (Ag-WT) (Figure 2a,b). Comparing the pre-symptomatic and symptomatic Tg2576 groups, there was a significant increase in intestinal mucus fucosylation in the symptomatic Tg2576 mice (Ag-Tg), when compared to the pre-symptomatic mice (Yg-Tg) (*p < 0.05,*
Figure 2b). We then assessed the expression levels of E-cadherin, an important apical tight junction protein with implications in intestinal epithelial homeostasis [41]. We detected a significant reduction in intestinal expression of E-cadherin in the pre-symptomatic Tg2576 mice (Yg-Tg), compared to age-matched WT littermates (Yg-WT) (*p < 0.05*, Figure 2c (top two rows),d). This reduction in E-cadherin expression did not persist in symptomatic Tg2576 mice (Ag-Tg) (Figure 2c (bottom two rows),d) We then performed FISH to examine the proximity of bacterial colonies and encroachment of their antigenic load onto the intestinal epithelium. We found widespread bacterial breach through the mucosal barrier in the intestines of pre-symptomatic Tg2576 mice (Yg-Tg), when compared to age-matched WT littermates (Figure 2e). Our data did not show the same level of bacterial breach at the symptomatic timepoint in Tg2576 mice (Ag-Tg) (Figure 2e). Bacterial breach was increased in Ag mice compared to Yg in both Tg2576 and WT littermates. Taken together, we found a significant impairment of IEB in the pre-symptomatic Tg2576 mice. 

### 2.3. Gut Microbiota Composition Is Significantly Different in Symptomatic Tg2576 Mice

The composition of gut microbiota can be influenced by intestinal epithelial dysfunction [42]. We performed 16S rRNA sequencing followed by qPCR to examine compositional differences in the gut microbiota of Tg2576 mice. Comparing the percentage abundance of Firmicutes and Bacteroidetes phyla (*F:B* ratio), we found no significant difference (1.3 vs. 1.5) in the pre-symptomatic Tg2576 mice (Yg-Tg) while the *F:B* ratio was significantly higher (13.4 vs. 1.6, *p* < 0.05) in the symptomatic mice (Ag-Tg), when compared to the age-matched WT littermates (Figure 3a). We then examined the bacterial diversity of gut microbiota in our samples. The alpha-diversity, or within-sample diversity, was not different at the pre-symptomatic timepoint, compared to WT littermates (Yg-Tg vs. Yg-WT). After visualization and analysis of alpha-diversity, or within-sample diversity, we observed no differences in OTUs between the groups (*p = 0.82*, Figure 3b). Upon visualization of beta-diversity, or between-sample diversity, with weighted UniFrac distances by principal coordinate analysis (PCoA), a significant clustering effect emerged along the PC1 axis (54.0% variation explained) at the symptomatic timepoint in Tg2576 (Ag-Tg), which was not observed at pre-symptomatic timepoint (Yg-Tg), when compared to age-matched WT littermates (Figure 3c). Examination of 16S data at the genus level showed a significant reduction in *Ruminiclostridium* in the pre-symptomatic Tg2576 mice (Yg-Tg), which persisted in the symptomatic Tg2576 mice (Ag-Tg), when compared to age-matched WT littermates (Figure 3d). Additionally, our data showed a significant increase in *Lactobacillus* abundance in the symptomatic Tg2576 mice (Ag-Tg), which was not present in the pre-symptomatic Tg2576 mice (Yg-Tg), when compared to age-matched WT littermates (Figure 3e). No significant shifts in the overall bacterial composition were observed at the order level in the pre-symptomatic Tg2576 mice (Yg-Tg), when compared to age-matched WT littermates (Appendix A). Our 16S data show that significant compositional differences exist in the symptomatic Tg2576 mice gut microbiota, which are not present in pre-symptomatic mice, when compared to WT littermate controls.

### 2.4. Plasma Levels of Inflammatory and Angiogenic Cytokines Are Elevated in Pre-Symptomatic Tg2576 Mice

To test the hypothesis that peripheral inflammatory events occur prior to the development of cerebral pathology, we profiled the circulatory cytokines in pre-symptomatic and symptomatic Tg2576 mice. The plasma levels of monocyte chemoattractant protein 1 (MCP-1; aka CCL2) were not significantly different in the pre-symptomatic Tg2576 mice (Yg-Tg), but were significantly elevated in the symptomatic mice (Ag-Tg), when compared to age-matched WT littermates (*p < 0.05,*
Figure 4a). Plasma levels of IL-9, VEGF-α, and IP-10 were elevated in the pre-symptomatic Tg2576 mice (Yg-Tg), compared to aged-matched WT littermate controls and symptomatic Tg2576 mice (*p < 0.001, p <* 0.001, respectively, Figure 4b–d). However, elevated IL-9, VEGF-α, and IP-10 plasma levels did not persist in the symptomatic Tg2576 mice (Ag-Tg), compared to aged-matched WT littermate controls (Figure 4b–d). These findings support the notion that peripheral disturbances in proinflammatory and angiogenic plasma cytokines occur before the presence of detectible deposition of cerebral Aβ in Tg2576 mice. 

### 2.5. Reduced Systemic Levels of Vitamin B12 Associated with Reduced Myelin Tight Junction Proteins in the CNS Is Present in Symptomatic Tg2576 Mice

The integrity of tight junctions is essential for proper neurochemical signal transduction of myelinated axons [43]. Brain immunohistochemistry (IHC) analysis showed a significant reduction in the expression levels of claudin 11, an intralamellar tight junction protein of internodal myelin in the CNS [44], in the symptomatic Tg2576 mice (Ag-Tg), which was not present in pre-symptomatic mice (Yg-Tg), when compared to age-matched WT littermates (Figure 5a,b). To explore possible gut-related mechanisms of myelin injury, we performed transcriptomic analysis of *cubilin,* an intestinal transporter that mediates B12 absorption in the ileum. Our data showed undetectable levels of cubilin mRNA expression in the ileum of pre-symptomatic Tg2576 mice (Yg-Tg), compared to the WT littermate controls (Yg-WT) (Figure 5c). In addition, we did not see any differences in cubilin mRNA expression at symptomatic timepoint (Ag-Tg), compared to age-matched WT littermate controls (Ag-WT). interestingly, we observed low levels of plasma B12 in the symptomatic Tg2576, when compared to age-matched WT littermates (Ag-WT) (*p* < 0.06, Figure 5d). These results support the notion that pre-symptomatic impairment of B12 intestinal absorptive function may be associated with long-term pathological effects on the CNS white matter integrity, characteristic of AD pathology.

### 2.6. Aβ Co-Localizes with the Intestinal Vasculature in the Pre-Symptomatic Tg2576 Mice

To determine whether Aβ deposits occur in the vasculature in the gut, we performed Aβ staining and two-photon microscopy at both timepoints. Our IHC data showed detectible intestinal Aβ accumulation that co-localized with vascular CD31 (aka PECAM1) in pre-symptomatic mice (Yg-Tg), which was not present in the age-matched WT littermates (Yg-WT) (Figure 6a, left column). This co-localization of intestinal Aβ deposits with vascular CD31 persisted in the symptomatic mice (Ag-Tg), when compared to age-matched WT littermates (Figure 6a, right column). There was also a significant increase in vascular-associated Aβ accumulation in the intestine of the symptomatic mice, when compared to the pre-symptomatic mice (Yg-Tg) (Figure 6a, bottom row). Using two-photon microscopy, we detected vascular-associated Aβ accumulation in the ileum and cecum, which was not associated with the presence of Aβ deposition in the hippocampus of the pre-symptomatic mice (Yg-Tg), when compared to age-matched WT littermates (Figure 6b, left column). We also detected vascular-associated Aβ accumulation in the ileum and cecum, which was associated with the presence of Aβ deposition in the hippocampus of the symptomatic mice (Ag-Tg), when compared to WT littermates (Figure 6b, right column). In addition, the Aβ antibody used also stained the luminal biofilms, denoting the availability of bacterial amyloid-like protein, in 15 months old age matched WT littermate control (Figure 6a, right column–Ag-WT) animals that were mostly observed in the luminal surface of gut epithelium. These findings support the hypothesis that the intestinal vascular-associated Aβ deposition occurs before the CNS vascular-associated Aβ deposition in Tg2576 mice.

### 2.7. Intestinal Aβ Deposition Is Present in Human Samples with AD Pathology

To validate the data obtained from our animal model, we obtained post-mortem autopsy samples from patients with known AD pathology. Five brain specimens (hippocampal region) and two corresponding intestinal autopsies from patients diagnosed with AD were examined and compared to non-AD controls (Appendix A). We first confirmed the presence of cerebral Aβ pathology in our AD samples compared to non-AD controls, using anti-β-amyloid 4G8 antibody staining (Figure 7a). We then performed the anti-β-amyloid 4G8 antibody staining on the intestinal autopsy samples from sporadic AD and non-AD patients. Our data showed the presence of Aβ deposits in the intestinal samples from AD patients, which was not present in the non-AD control (Figure 7b). These results support the hypothesis that intestinal Aβ deposits exist in human AD pathology. 

## 3. Discussion

An emerging theme in studies of various NDDs is the role of peripherally initiated dysfunction that propagates centrally to promote cerebral pathology [18,19,20,21,22,45,46] Here, we showed the presence of dysregulation of IEB and intestinal absorption as well as increased inflammatory and angiogenic plasma cytokines prior to development of cerebral Aβ accumulation in Tg2576 mouse model of AD. Our findings support the notion that Aβ-induced vascular injury may begin in the intestinal vasculature and lead to loss of IEB integrity as assessed by a reduction in mucus and E-cadherin production prior to cerebral pathology (Figure 8). This loss of barrier function was associated with increased plasma levels of inflammatory and angiogenic cytokines including IL-9, VEGF-α, and IP-10. In addition to the loss of IEB integrity, here we report a significant reduction in B12 transporter gene expression, cubilin, at the pre-symptomatic timepoint followed by reduced levels of plasma B12 at the symptomatic timepoint in Tg2576 mouse model of AD prior to detection of myelin-related white matter injury in the CNS. Lastly, we showed that Aβ deposition is present in intestinal autopsy specimens obtained from AD patients. Our findings support the hypothesis that intestinal dysregulation can induce a pro-inflammatory state in the peripheral circulation which may initiate or contribute to the progression of cerebral Aβ accumulation in a genetically susceptible host. 

Pre-clinical models remain an important tool for mechanistic examination of pathogenesis and progression of NDDs. One of the most widely used pre-clinical models of amyloidopathy is the Tg2576 mouse model, known to exhibit cerebral Aβ amyloidosis pathology beginning around 9 months [31,34]. To determine whether gut-centric dysregulation precedes the onset of Aβ pathology in the CNS, we examined gut homeostasis by measuring mucus barrier integrity, the proximity of bacterial populations to the intestinal epithelium, and expression levels of tight junction proteins on the apical surface of intestinal epithelial cells (IECs) [47]. Our data showed a significant presence of disease-like state at pre-symptomatic timepoints, identified by attenuation of mucus fucosylation, bacterial colonies residing in close proximity to the epithelial cells, and reduced expression of E-cadherin tight junctions at the apical surface of IECs. Additionally, our IHC and two-photon imaging data suggest that the intestinal vascular-associated Aβ deposition in pre-symptomatic animals may be the underlying mechanism of IEB injuries. Our findings support the hypothesis that gut dysfunction may occur prior to onset of cerebral amyloidopathy in AD or other cerebral proteinopathies. Our study lends support to the previous studies in germ-free animals, gnotobiotic mice (with known microbiota), and fecal microbiota transplantation (FMT) in AD transgenic mouse models that suggested a role for the gut homeostasis in amyloid-related pathogenesis [21,22].

Loss of bacterial diversity, a commonly used marker of dysbiosis (defined as shift in gut microbiome composition that is accompanied by disease-like events [48]) is associated with multiple diseases such as inflammatory bowel disease (IBD) [49], obesity and metabolic syndrome [50], and hospital-acquired pseudomembranous colitis [51]. Loss of gut microbiota diversity has been reported in AD patients [52,53]. Our data showed a significantly reduced alpha- (within-sample) diversity as well as a significantly different beta- (between-sample) diversity at the symptomatic timepoint, which were not present pre-symptomatically, when compared to age-matched controls. Our observation suggests that changes in the composition of gut microbiota may be a secondary effect to Aβ-induced injuries at the IEB and in the brain. In addition to bacterial diversity, the Firmicutes:Bacteroidetes (F:B) ratio is reported in the microbiome field as a measure of compositional changes in the gut microbiota. Consistent with studies showing reduced *Bacteroidetes* in IBD and other conditions with IEB dysregulation [54], our study also showed a significant loss of *Bacteroidetes* population at the symptomatic timepoint, which was not present pre-symptomatically. Decreased levels of *Ruminiclostridium* genus, a major producer of short-chain fatty acids, have been previously reported in IEB dysfunction [55] and in diabetic patients [56]. Consistently, our data showed that genus *Ruminiclostridium* was significantly reduced in the pre-symptomatic mice with IEB dysfunction. Together, our examination of microbial diversity and phylum-level abundance levels argue against the notion that the loss of microbial diversity leads to IEB dysfunction. Instead, our data suggest that IEB dysfunction may be responsible for the observed changes in the composition of gut microbiota. 

Accumulating evidence has begun to shift our understanding of dynamic microbiota-gut-immune interactions in health and disease [57], and the disturbances in plasma cytokine profile is often reported as the downstream effect of these complex interactions in the gut. We found that MCP-1 plasma levels were only elevated at the symptomatic timepoint, and not pre-symptomatically. MCP-1 is involved in the migration and activation of microglia or macrophages as a secondary response to the presence of Aβ plaques or demyelinating injury [58,59,60] and our finding supports this notion. 

IL-9 plays a role in the regulation of the adaptive immune response to goblet cell and Paneth cell pathologies and has been shown to reduce tissue repair capability in the gut after injury [61,62]. We found that IEB dysfunction in pre-symptomatic mice was associated with elevated levels of plasma IL-9, suggesting that IL-9 mediated modulation of gut immunity could be a key contributor to IEB dysregulation. The role of IL-9 at the blood brain barrier has not been clearly elucidated. Elevated levels of IL-9 cytokines in the cerebrospinal fluid (CSF) samples from AD patients has been reported to be correlated with AD [63], suggesting a possible “protective” role of IL-9 in the CSF. Future studies are needed to evaluate the role of IL-9 in plasma, cerebrospinal fluids, or lymphatics and its contribution to AD pathogenesis.

It has been shown that VEGF-α, a potent angiogenic factor, is upregulated in the AD brain, speculated to be a secondary response to Aβ-induced vascular injuries in the brain [64]. Our findings, however, suggest an alternative possibility. We found elevated plasma levels of VEGF-α in the pre-symptomatic animals with IEB dysfunction. This suggests that IEB injury, in the absence of cerebral Aβ, may cause VEGF-α increase in the plasma that can in turn lead to its elevated levels in the brain.

Increased levels of IP-10, an angiogenic and inflammatory cytokine [65], have been reported in CSF samples of patients with mild cognitive impairment (MCI) and mild AD but not in severe AD [66]. However, results of other studies examining IP-10 levels in the CSF and plasma samples from AD patients have been inconsistent [67]. 

Our results show that elevated plasma levels of IP-10 was present in the pre-symptomatic mice with IEB dysfunction. Considering the abovementioned human data [66] and our observations in the Tg2576 animal model, we speculate that IP-10 levels may be elevated during the early stages AD pathogenesis, yet reduced in advanced disease. Future studies are needed to elucidate the specific inflammatory events related to changes at the IEB, in the circulation, and in the brain in the course of AD pathogenesis.

Neuroimaging studies have demonstrated the presence of white matter injury in the human AD brain [68]. Additionally, a recent single-cell transcriptomic study revealed that myelination-related pathways were perturbed in the human AD brain [69]. To assess white matter injury, we examined claudin-11, an essential constituent of tight junctions of myelin, involved in maintaining the electrophysiological homeostasis and signal transmission of myelinated neurons [44,70,71,72,73,74]. We found that claudin-11 (measure of myelin basic protein) expression levels were significantly reduced in the brain at the symptomatic timepoint, implicating a claudin-11-mediated molecular mechanism for white-matter injury seen in AD. Future studies are needed to evaluate the role of claudin-11 in the development of white-matter disease, characteristic of human AD. 

In support of recent findings in other neurodegenerative diseases [45], the strong positive Aβ antibody staining in the luminal side of the intestinal epithelium (Figure 6a) supports the notion that bacterial amyloid may be one of the major source of luminal biofilm [75,76,77,78], as also hypothesized by others. In addition, the presence of relatively less robust yet clearly positive Aβ staining on the lamina propria side of the intestinal epithelium suggests a concentration gradient from the gut lumen towards the sub-epithelial vasculature in AD. Our observation of Aβ co-localized with the vascular maker, CD31, lends further support to the possibility of bacterial origin of amyloid fibrils. These translocated amyloid fibrils, possibly due to the dysfunctional gut barrier integrity, can lead to local and systemic changes in the immune response which can contribute to the initiation or progression of cerebral amyloidopathy observed only in AD animal model. 

Interestingly, we also identified an impairment of the intestinal absorptive function in Tg2576 mice that may be linked to white matter disease in this AD mouse model. We found a significant loss of *cubilin*, also known as intestinal intrinsic factor–cobalamin receptor, mRNA expression in the ileum of pre-symptomatic Tg2576 mice. Intestinal cubilin is the primary transporter of vitamin B12 (B12). Biosynthesis of B12 is confined to microbial fermentation by a specific bacteria and archaea [79]. Once bound to gastric intrinsic factor (IF), the IF-B12 complex must be absorbed via the intestinal cubilin transporters [33]. In line with the evidence of intestinal dysfunction, we observed low levels of *cubilin*, the intestinal transporter of B12 in pre-symptomatic mice that precedes low levels of circulating B12 in blood plasma of symptomatic Tg2576 mice compared to the WT littermate controls. Cubilin expression is known to reduce with inflammation and therefore we assume that gut dysfunction leads to reduced cubilin expression as observed in Tg2576 pre-symptomatic mice [80]. B12 plays an essential role in myelin homeostasis [81,82]. B12 deficiency is associated with an increased risk of dementia [83] and a wide range of gastrointestinal and neurological diseases [82,84]. We cautiously speculate that loss of *cubilin* expression in pre-symptomatic mice with genetic pre-disposition for cerebral Aβ pathology leads to reduced B12 levels that may contribute to long-term loss of myelin integrity observed in symptomatic animals [81,83]. In our opinion, intestinal homeostasis shifts toward overcompensating for the loss of gut integrity in early stages, as evident by a moderate increase in mucus fucosylation, E-cadherin, and *cubilin* at the symptomatic timepoint, when compared to the pre-symptomatic group. Importantly, however, the gut barrier remains impaired at the symptomatic timepoint when compared to age-matched WT controls. In line with the gut dysfunction, we observed reduced levels of circulating B12 at pre-symptomatic compared to symptomatic timepoint. Taken together, our data strongly suggest that the gut dysfunction precedes cerebral Aβ pathology and reduced white matter integrity. Mechanistic studies using knock out models of Cubilin are required to test this hypothesis.

Lastly, we used human samples obtained from AD patients to examine the presence of gut dysfunction in AD pathology. We first confirmed the presence of cerebral Aβ deposition in all available samples. Importantly, and in support of our hypothesis, we detected Aβ aggregation on the apical side of intestinal epithelium in gut samples from patients diagnosed with AD. Overall, our findings from Tg2576 mouse model, complemented by the detection of Aβ aggregates in the intestinal samples of AD patients, strongly support our hypothesis that gut dysfunction occurs in AD pathology and may precede the development of CNS pathology. 

## 4. Conclusions

Disturbances in gut physiology may influence the risk of AD and its progression. Fortunately, advances in non-invasive clinical imaging, examination of intestinal health, and affordable microbiome sequencing technologies allow for near-future integration of gut health into the clinical management of AD patients. If confirmed by future pre-clinical and clinical studies, the impact of this work will be significant if gut dysfunction can be detected before the clinical manifestations of AD. Future therapeutic strategies to reverse pathology in AD may involve early manipulation of gut physiology and its microbiota. Mechanistic studies to evaluate molecular pathways involved in modulation of the peripheral immune system and a rigorous analysis of the cognitive phenotype of AD models will be of high interest to the field. 

## 5. Methods

### 5.1. Mice

The Tg2576 transgenic mouse model of AD was used in this study. These mice overexpress the 695-amino acid isoform of human Alzheimer beta-amyloid (Aβ) precursor protein (APP_695_) containing a double mutation of Lys670Asn, Met671Leu (i.e., Swedish mutation) under the control of the hamster prion protein (PrP) promoter [85], resulting in elevated levels of Aβ aggregates, extensive cerebral amyloid pathology, and cognitive deficits [31]. These animals develop parenchymal plaques beginning at around 9 months of age with some vascular amyloid, which eventually becomes widespread in the cortical and limbic structures by around 16 months of age [34].

The study was conducted under protocols approved (19 February 2020) by the Center for Laboratory Animal Medicine and Care (CLAMC) (Protocol number AWC-18-0127 and AWC-18-0132) at the University of Texas McGovern Medical School. We formed the breeding pairs with young (2 months old) Tg2576 male (Tg(APPSWE)2576Kha N20+?) mice (model 1349) and young Balb/c female mice that were purchased from Taconic. Prior to breeding, all the animals purchased were housed for one month to normalize their microbiota and adapt to the new environment. The breeding pairs were used to generate F1 generation followed by genotyping to confirm the presence of the inherited mutation. Tg2576 and WT littermate mice obtained from F1 generation were used in the study. All the experiments shown here were performed with animals obtained from F1 generation. The pups were weaned 21 days after birth and both sexes were used in the study. Mice were housed up to 5 per cage (individually ventilated, changed weekly or bi-weekly under HEPA-filtered workstations) in standard facilities with a 12-h light/dark schedule (lights on at 7AM) in a temperature- (21.7–22.8 C) and humidity-controlled (40–60 RH) controlled vivarium, with ad libitum access to food (LabDiet 5053 and 5058, pelleted, irradiated at manufacturer, stored at room temperature for up to 6 months) and water (filtered tap water, pH 6–8, not acidified nor chlorinated). 

### 5.2. Immunohistochemistry (IHC) and Microscopy

IHC was performed as previously described in [37,40]. Formalin-fixed, paraffin-embedded 5μm-thick intestinal sections were used for the following staining protocols. For lectin staining, terminal mucin glycans were examined using a panel of FITC-conjugated lectins: *Ulex europaeus* agglutinin-1 (UEA-1) for terminal fucose. De-paraffinized sections were incubated with citrate buffer pH 6 (Vector Labs) for 20 min in a pressure cooker and blocked with PBS containing 10% BSA. Sections were then stained in a humidified chamber with FITC-labeled lectin (10 μg/mL) for 1 h at room temperature (RT). Sections were washed with PBS, counterstained with Hoechst for 10 min at RT and mounted using aqueous mounting media (Sigma Aldrich). 

For all other gut staining protocols, after the dewaxing and rehydration steps, a heat-inducing antigen retrieval procedure using 1 mM EDTA buffer at pH 8.0 was performed, with subsequent washing in PBS. After a blocking step, sections were incubated with primary antibodies (rat monoclonal anti-E-cadherin (Cat #: ab11512, abcam); beta-Amyloid (Aβ) –rabbit polyclonal beta amyloid (Cat #: ab2539, abcam); rat anti-CD31 (PECAM-1, Cat #: 550274, BD Bioscience)) overnight, followed by incubation with fluorescent secondary or HRP immunoglobulins (1:1000, multiple routinely used vendors) for 60 min. Immunoreactions were developed with diaminobenzidine (DAB) diluted in DAB Substrate Buffer (Peroxidase chromogen or Substrate solution from N-Histofine DAB-2 V, Tokyo, Japan). The sections were counterstained with hematoxylin for IHC or DAPI (ThermoFischer) for f-IHC prior to visualization. Images were obtained on a Leica DM8 i SPE confocal microscope.

Brain sections fixed in tissue freezing media were cut with 15μm thickness. Thioflavin S (MilliporeSigma, Mfr. No. T1892-25 G) staining for detection of amyloid-β (Aβ) plaques in the brain sections was performed by incubating the sections with 1% thioflavin-S in 0.01 mol/L PBS in the dark for 8 min at 22 °C, followed by washing in 70% ethanol for 3 min. Claudin-11 antibody (Abcam, ab53041) was used to stain fresh brain sections for oligodendroglia specific tight junction protein expression. Purified anti-β-amyloid, 17–24 Antibody (Biolegend, clone 4 G8, previously Covance catalog # SIG-39220) was used to stain 6-μm thick, paraffin-embedded human brain and anti-Amyloid antibody, while β 1–40/42 (Millipore Sigma, AB5076) was used to stain 5-μm thick, paraffin-embedded gut autopsy sections obtained from AD patients and human controls. 

### 5.3. Fluorescent In Situ Hybridization (FISH)

Formalin-fixed paraffin-embedded intestinal tissue sections (5 µm) were initially treated with lysis buffer for 1 h at 37 °C and then hybridized at 51 °C with a 20 bp bacteria-specific probe (EUB 338: GCTGCCTCCCGTAGGAGT) to visualize and quantify the proximity of luminal bacterial colonies and the corresponding antigen load on the epithelium. Following the overnight hybridization, the intestinal sections were counter-stained with 4,6-diamidino-2-phenylindole (DAPI, Dihydrochloride, Cat #: D1306 by Invitrogen) as previously described [40] for visualization of cell nuclei. Images were obtained and analyzed using a Leica DM8 i SPE confocal microscope.

### 5.4. Sample Collection and 16 S rRNA Sequencing

Intestinal luminal content was collected from mice at the time of tissue harvest and stored in sterile tubes at −80 °C until being analyzed. The bacteria taxa in each sample were analyzed by amplifying the V4 variable region of the 16S ribosomal RNA (rRNA) gene using high-throughput sequence analysis (Illumina MisSeq platform; Illumina, San Diego, CA) [86]. Quality filtered 16S rRNA sequences were clustered into operational taxonomic units (OTUs), with 97% similarity, by closed reference OTU-picking using the UCLUST algorithm and GreenGenes reference database (v13.5) as implemented in Quantitative Insights Into Microbial Ecology (QIIME versions 1.6 and 1.7) [87,88,89]. Sequences were checked for chimeras using ChimeraSlayer with standard options as implemented in QIIME. Sequences not clustered were identified using the Ribosomal Database Project to the lowest possible taxonomic level [90]. The data were randomly rarefied to 10,000 sequences per sample before any downstream analysis. *ATIMA* (Agile Toolkit for Incisive Microbial Analyses) developed by Alkek Center for Metagenomics and Microbiome Research (CMMR) at Baylor College of Medicine was used for analysis and visualization of microbiome data sets. 

### 5.5. Cytokine Measurements

Blood was collected by cardiac puncture before perfusion and centrifuged (6000× *g* for 10 min at 4 °C) and plasma was collected and stored frozen (at −80 °C) until use. Changes in circulating inflammatory cytokines were examined by Multiplex assay kits (purchased from Millipore Sigma MILLIPLEX Mouse Cytokine or Chemokine Panel 1 Custom Premixed Panel targeting Eotaxin, G-CSF, GM-CSF, IFN-γ, IL-1α, IL-1β, IL-2, IL-3, IL-4, IL-5, IL-6, IL-7, IL-9, IL-10, IL-12 p40, IL-12 p70, IL-13, IL-15, IL-17 A, CXCL10 aka IP-10, KC, CCL2 aka MCP-1, M-CSF, CXCL9 aka MIG, CCL3 aka MIP-1α, MIP-1β, MIP-2, RANTES, TNF-α, VEGF-α) according to the manufacturer’s instructions. Briefly, 25 µl of plasma samples collected from each mouse were thawed completely and diluted with the same amount of Assay Buffer provided in the kits. The assays were performed blindly and in duplicates. The reports generated by MILLIPLEX^®^ Analyst 5.1 Software were carefully reviewed and only cytokines levels above the detection limit and below the saturated value were considered. The detection limits for the aforementioned cytokines were between 10,000 pg/mL and 3.20 pg/mL, respectively. Cytokine abbreviations: granulocyte-colony stimulating factor (G-CSF), granulocyte-macrophage-colony stimulating factor (GM-CSF), interferon gamma (IFNg), interleukin- (IL-), C-X-C motif chemokine 10 (CXCL10) aka interferon gamma-induced protein 10 (IP-10), KC, chemokine (C-C motif) ligand 2 (CCL2) aka monocyte chemoattractant protein 1 (MCP-1), macrophage-colony stimulating factor (M-CSF), CXCL9 aka Monokine induced by gamma interferon (MIG), CCL3 aka macrophage inflammatory protein 1-alpha (MIP-1α), MIP-1β, MIP-2, RANTES (regulated on activation, normal T cell expressed and secreted, tumor necrosis factor-alpha (TNF-α), vascular endothelial growth factor-alpha (VEGF-α)

### 5.6. mRNA Gene Expression

To quantify relative mRNA expression levels of IL-6, cubilin and glyceraldehyde 3-phosphate dehydrogenase (GAPDH) mRNA was extracted from intestinal mucosa samples using the miRNeasy^®^ mini kit (QIAGEN). Then, 1.0 µg of RNA was reverse-transcribed to single-stranded cDNA using the RevertAid H minus First Strand cDNA Synthesis Kit (Thermo Fischer, USA). Reverse transcriptase real-time (RT) PCR was performed using the Quant Studio 3 Real-Time PCR system (Applied Biosystems, USA). The RT-PCR reaction mix (adjusted with H_2_ O to a total volume of 20 µl) contained 2 µl template DNA, 10 µl TaqMan Fast advanced master mix (Thermo Fischer, USA), 0.5 µl of the TaqMan assay (Cubn: Mm01325077_m1, Hif1 a: Mm00468869_m1 and Gapdh: Mm99999915_g1 from Thermo Fischer, USA). Relative mRNA target gene expression levels (Ratio = [(E_target_) ^dCPtarget (control-sample)^] **/** [(E_ref._) ^dCPref. (control-sample)^]) were normalized to the house keeping gene glyceraldehyde 3-phosphate dehydrogenase (GAPDH) and used as a reference. Subsequently, intestinal mucosal cytokine of the WT littermate control group at 6 months (Yg-WT) was set to 1.0 and used as the calibrator to identify the relative mRNA fold difference between the WT littermate controls and Tg2576 mice at 6 and 15 months.

### 5.7. Vitamin B12 Measurement by Mass Spectrometry

Blood was collected by cardiac puncture before perfusion and centrifuged (6000× *g* for 10 min at 4 °C) and plasma was collected and stored frozen (at −80 °C) until use. Changes in circulating vitamin B12 were examined by mass spectrometry. Vitamin B12 was extracted from plasma using liquid-liquid extraction as described in earlier publications [91,92,93]. The HPLC column used was Zorbax eclipse XDB C-18, 1.8 micron, 4.6 × 100 mm Agilent technologies Santa Clara, CA, USA). The mobile phases used were 0.1% formic acid in water and acetonitrile as mobile phase A and B, respectively. The initial gradient started with 2% of B and maintained until 4 min, and it was ramped in a linear fashion to 30% of B until 6.5 min, at 7 min 90% B, 12 min 95% B, at 13 min 2%B and reequilibration started until the end of gradient at 20 min. The flow rate used was 0.2 mL/min. The injection volume was 10 µL. Data were acquired using Agilent Mass hunter acquisition software and analyzed using Agilent quantitative analysis software.

### 5.8. Two-Photon Microscopy to Visualize Amyloid-β in the Brain and Gut

A two-photon microscopy system (Bruker Ultima Investigator) equipped with a laser source (InSight DeepSee, 680–1300 nm and 1040 nm, Spectra Physics) was employed to perform ex vivo imaging of samples. Two-photon excitation of methoxy-X04 (10 mg/kg, i.p., Abcam ab142818, USA) and DyLight649 labeled lectin from Lycopersicon esculentum (Vector DL-1178) was performed at 810 nm and each emission signal was separately detected by two GaAsp detectors and digitally recorded for image analysis. Methoxy X04 was i.p. injected 24 h prior to the imaging. Mice under anesthesia received an i.v. injection of DyLight649 labeled lectin through the jugular vein 5 min prior to sacrifice for ex vivo imaging. Tissues including the brain, ileum, and cecum were collected followed by sample fixation in 4% PFA. After a couple of washes with 1 xpbs, samples were mounted inside a 7 cm petri dish with 1× pbs. The meningeal vasculature in the parietal cortex was imaged from the pial surface of the region, hippocampal vasculature was imaged in the sagittally mounted brain tissues, ileum and cecum were imaged from luminal surface of each tissues cut and mounted to expose the lumen. All regions were imaged 2–3 times per tissue and Vision 4 D software (Arivis, DC, USA) was used to analyze the image data.

### 5.9. Statistics

Data were tested for normal distribution using the Kolmogorov–Smirnov test. Normally distributed data are presented as means with standard error while the medians with their range are given for non-normally distributed data. Significance of differences between Yg-WT (age-matched littermate controls for pre-symptomatic Tg2576), Yg-Tg (pre-symptomatic Tg2576), Ag-WT (age-matched littermate controls for symptomatic Tg2576) and Ag-Tg (symptomaticTg2576) mice were analyzed using the two-way ANOVA of variance test, followed by either Bonferroni/Tukey or Sidak’s multiple comparison post-hoc tests. Differences between the Yg-WT and Yg-Tg group were tested using Student’s t-test followed by the Mann–Whitney test for non-normally distributed data. All the experiments are performed blinded. Differences between the groups were considered significant at * *p* < 0.05, ** *p* < 0.01, *** *p* < 0.001. SPSS 16.0 (IBM, USA) for Windows 7 was used for data analysis. Prism 5.0 software (Graph Pad Software, Inc., La Jolla, CA, USA) for Windows, was used for data presentation and also for data analysis. All investigators were blinded to genotype. 

## Figures and Tables

**Figure 1 ijms-21-01711-f001:**
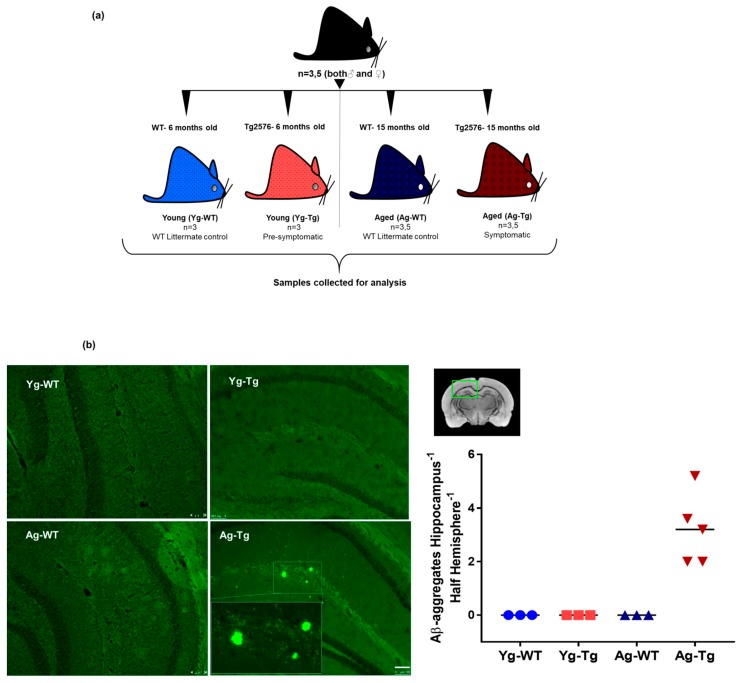
Experimental design, timeline, and presence of cerebral Aβ pathology in Tg2576 mouse model. (**a**) Associated timeline in our animal model of pre-symptomatic at 6 months (Yg-Tg) and symptomatic at 15 months (Ag-Tg) timepoints. (**b**) Thioflavin S staining in Tg2576 mice for visualization of parenchymal Aβ plaques in coronal brain slices confirms the absence of Thioflavin S-positive plaques at 6 months and their presence in the subiculum and hippocampal formation at 15 months. *n* = 4 (or) 5. Data are expressed as mean +  SEM, as well as individual values, and are obtained from >two independent experiments. Magnification 10X; section thickness- 15 μm. Scale bars: 100 μm WT wild-type, Tg-Transgenic; Green dots- Aβ plaques.

**Figure 2 ijms-21-01711-f002:**
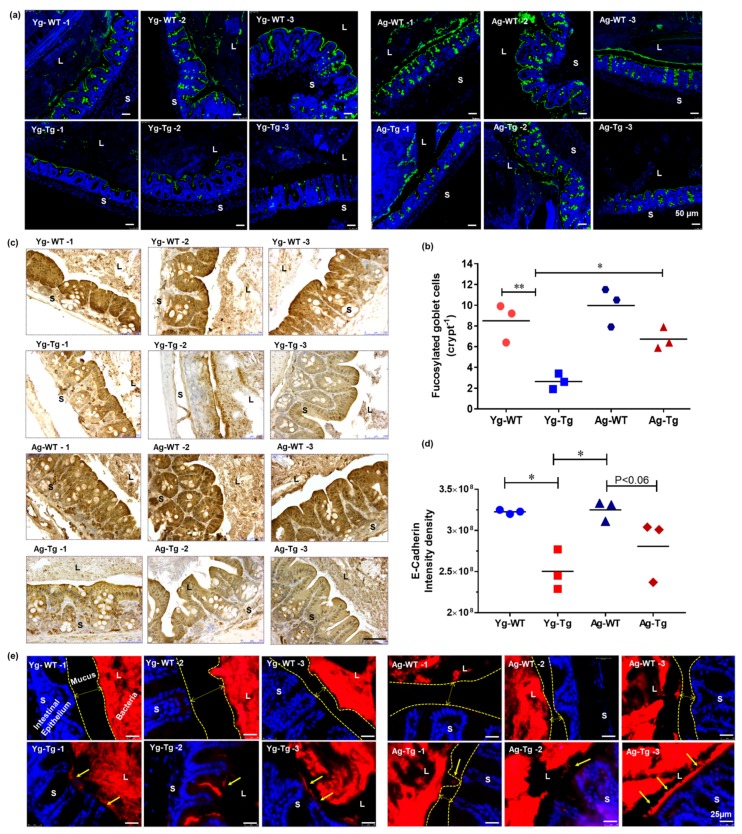
Gut dysfunction occurs before development of Aβ pathology in the brain in Tg2576 mice. (**a**,**b**) Lectin staining and mucus fucosylation shows a significant reduction in mucus fucosylation with *Ulex europaeus* agglutinin staining of the terminal mucus fucose in the cecum of Tg2576 at 6 months when compared to age-matched WT controls. (**c**,**d**) Immunohistochemical staining of intestinal epithelial shows a significant reduction in E-cadherin expression in Tg2576 mice when compared to WT littermate controls at 6 months. (**e**) Widespread bacterial breach through the mucosal barrier and the corresponding antigenic load onto the intestinal epithelium detected by FISH in the cecum of Tg2576 mice at 6 months. *n* = 3 per group. Data are expressed as mean ± SEM, as well as individual values, and are obtained from >two independent experiments at various times. ** p < 0.05, ** p < 0.01. P* values were calculated using Two-Way ANOVA analysis with Tukey’s multiple comparisons test (**b**) and (**d**). Scale bars: 50 μm (**a**), 250 μm (**c**), 25 μm (**e**). WT wild-type, Tg-Transgenic. Green- mucus (a), brown- e cadherin (**c**) and red- bacteria (**e**), blue-DAPI nuclei.

**Figure 3 ijms-21-01711-f003:**
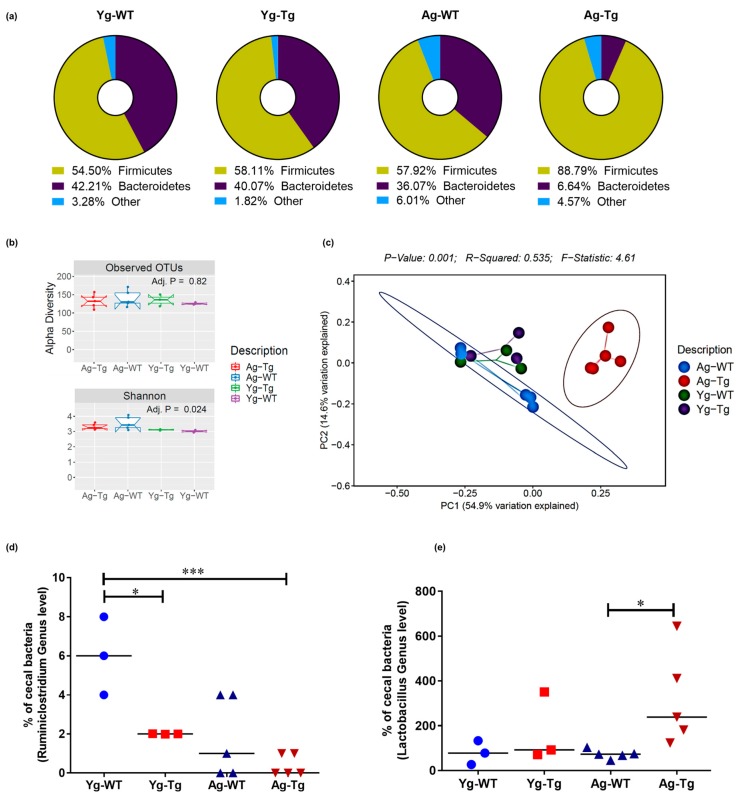
Compositional differences in gut microbiota by 16S rRNA sequencing of intestinal luminal content. (**a**) Average enrichment of top bacterial phyla of Firmicutes and Bacteroidetes in Tg2576 vs. WT littermate controls by qPCR. (**b**) Visualization of beta-diversity, or between-sample diversity, with weighted UniFrac distances by principal coordinate analysis (PCoA) shows a clustering effect by strain between Tg2576 and WT littermate controls at 15 months, which is not observed at 6 months. (**c**) Number of observed Operational Taxonomic Units (OTUs) and the alpha-diversity, or within-sample diversity (Shannon diversity index) when comparing Tg2576 and WT littermate groups. (**d**) Differences at the genus level show a significantly reduced abundance of *Ruminiclostridium* in the cecal content of Tg2576 mice starting at 6 months and persisting at 15 months compared to age-matched WT littermate controls. (**e**) Differences at the genus level show a significant increase in *Lactobacillus* abundance in Tg2576 mice at 15 months, which was not present at 6 months, when compared to age-matched WT controls. *n* = 3,5 per group. Data are expressed as mean ± SEM, as well as individual values, and are obtained from >2 independent experiments at various times. ** p < 0.05, *** p < 0.001. P* values were calculated using Two-Way ANOVA analysis with Sidak’s multiple comparisons correction (**d**,**e**).

**Figure 4 ijms-21-01711-f004:**
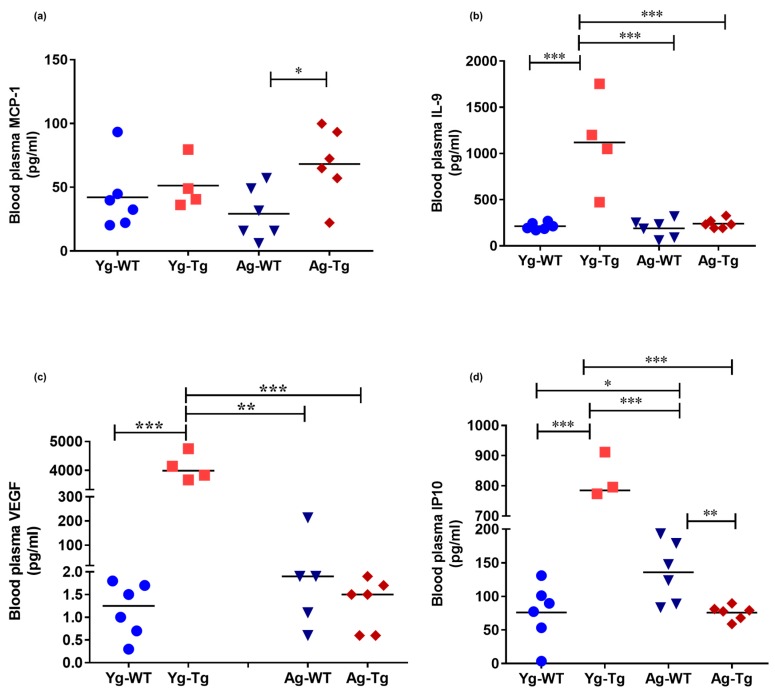
Elevated levels of inflammatory and angiogenic cytokines in the peripheral circulation of pre-symptomatic Tg2576 mice. (**a**) Plasma levels of MCP-1 are significantly elevated at 15 months when cerebral Aβ pathology is detectible but not at the pre-symptomatic timepoint of 6 months (Yg-Tg) in Tg2576 mice. (**b**–**d**) Plasma levels of IL-9, VEGF-α, and IP-10 are elevated in the plasma at 6 months in Tg2576 mice (Yg-Tg), prior to any detectible Aβ in the brain. IL-9, VEGF-α, and IP-10 plasma levels are not significantly elevated at 15 months when significant Aβ pathology exists in the Tg2576 mice (Ag-Tg) brains. *n* = 4,6 per group. Data are expressed as mean ± SEM, as well as individual values, and are obtained from three independent experiments at various times. ** p* < 0.05, *** p* < 0.01, and *** *p* < 0.001. *P* values were calculated using Two-Way ANOVA with Sidak’s multiple comparisons correction (**a**–**d**).

**Figure 5 ijms-21-01711-f005:**
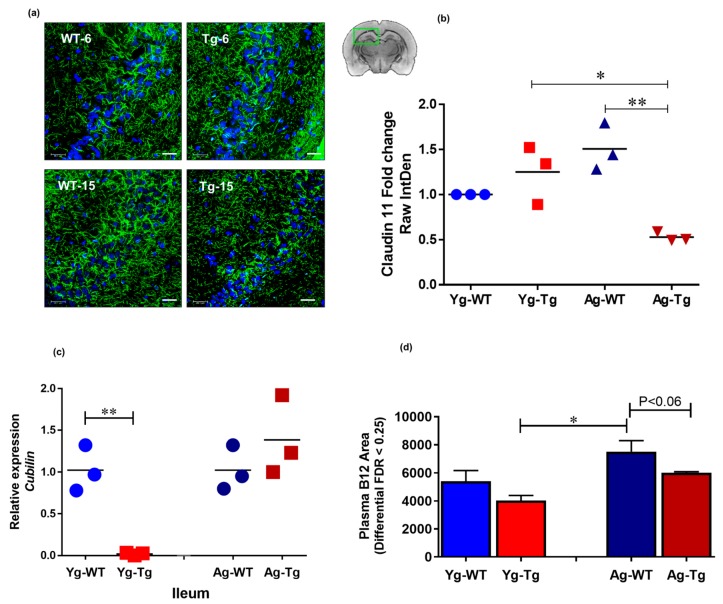
Claudin-11 expression is significantly decreased in symptomatic Tg2576. (**a**,**b**) immunohistochemical staining of oligodendroglia-specific claudin-11 shows significant loss of claudin-11 expression in Tg2576 mice at 15 months (Ag-Tg), but not at 6 months (Yg-Tg). (**c**) Transcriptomic analysis using Taqman probes targeting *cubilin* in the ileum of Tg2576 and WT littermate controls shows nearly undetectable Cubilin gene expression in the ileum of Tg2576 mice at 6 months (Yg-Tg) compared to the WT littermate controls (Yg-WT). (**d**) Metabolomics analysis using mass spectrometry targeting B12 in the blood plasma of Tg2576 and WT littermate controls shows low levels of B12 in the plasma of Tg2576 mice at 15 months (Ag-Tg) compared to the age-matched WT littermate controls (Ag-WT) (*p* < 0.06). (**a**) Green: Claudin-11, Blue: DAPI nuclei. Magnification 60×. Bar indicates 25 µm. *n* = 3 per group. Data are expressed as mean ± SEM, as well as individual values, and are obtained from >two independent experiments at various times. ** p* < 0.05, ** *p* < 0.01, *P* values were calculated using Two-Way ANOVA analysis with Tukey’s multiple comparisons test (**b**–**d**).

**Figure 6 ijms-21-01711-f006:**
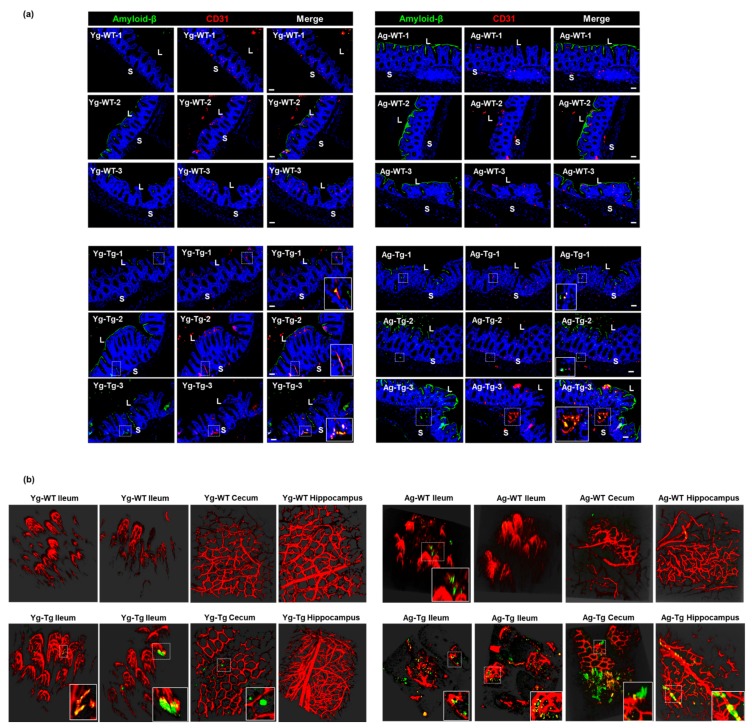
Aβ co-localization with the intestinal vasculature in pre-symptomatic Tg2576 (Yg-Tg). (**a**) Immunohistochemical staining of Aβ by 4G8 antibody shows detectible Aβ deposits co-localized with vascular CD31 in the intestinal samples from Tg2576 at pre-symptomatic timepoint of 6 months (Yg-Tg) compared to age-matched WT littermate controls (Yg-WT) which significantly intensifies by 15 months (Ag-Tg). (**b**) Two-photon imaging of intestinal tissue shows detectible levels of Aβ accumulation with ileal and cecal vasculature in the gut but not in the hippocampus of the brain at 6 months in Tg2576 mice (Yg-Tg). Ileum, cecum and hippocampus samples show strong Aβ plaques co-localized with intestinal and cerebral vasculature at 15 months in Tg2576 mice (Ag-Tg). Data are obtained from >two independent experiments per group at various times (**a**) and separate set of two independent experiments at different times (**b**) due to dye injection. (**a**) Magnification 20×. Bar indicates 50 µm. L—luminal, S—serosal, Yg—young, Ag—aged, Tg—transgenic, WT—wild type. Red—CD31, Green—amyloid β, Blue—DAPI nuclei. **(b)** Red—vasculature, Green—amyloid β. (**b**) Magnification 20×.

**Figure 7 ijms-21-01711-f007:**
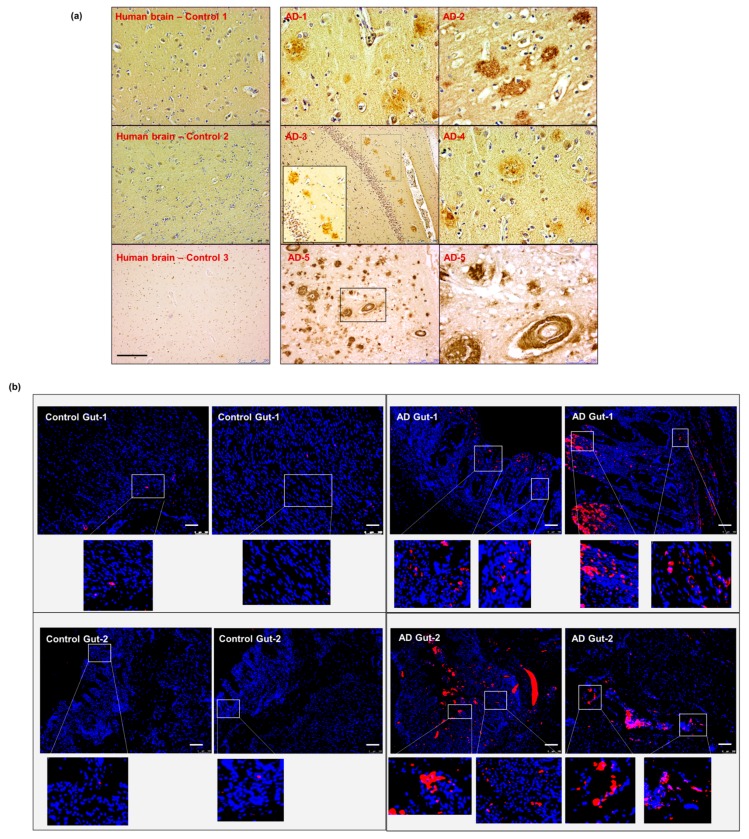
Aβ detection in human sporadic AD gut and brain samples. (**a**) All five AD human brain samples show significant levels of parenchymal and perivascular Aβ depositions. (**b**) The gut tissues obtained from two of the same set of AD patients (corresponding to brain samples AD-1 and AD-2 in panel (**a**) show the presence of Aβ-aggregation on the epithelium and in the intestinal mucosa. Scale indicates 250 µm (**a**) and 100 µm (**b**). Magnification 20× (**a**). AD—Alzheimer Disease brain. Brown—amyloid β aggregates. (**b**) AD—Alzheimer Disease gut. Magnification 20× (**b**). Red—amyloid β, Blue—DAPI nuclei.

**Figure 8 ijms-21-01711-f008:**
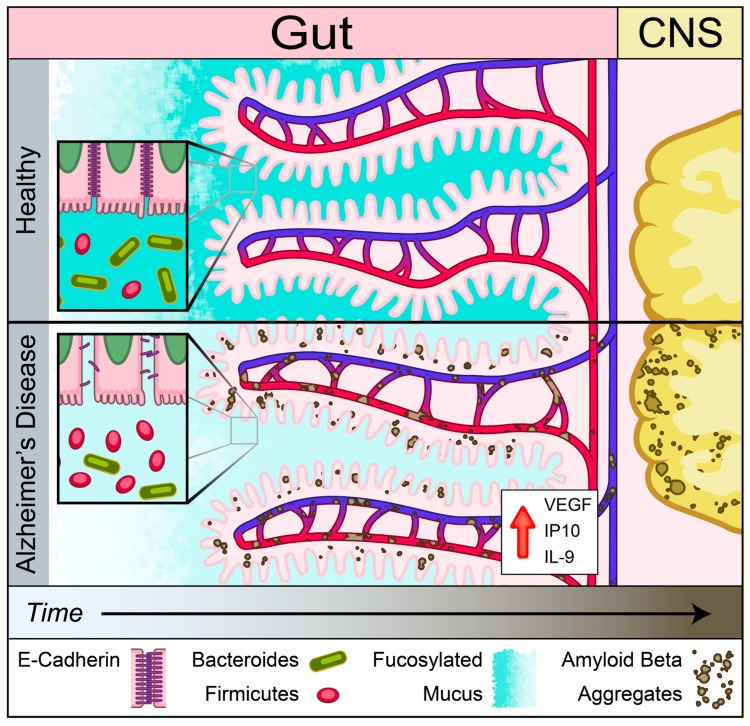
Schematic of gut–brain axis changes in Tg2576 mouse model of AD. Gut precedes in Aβ pathology compared to brain identified using Tg2576 AD mouse model. Reduced mucus fucosylation, reduced e-cadherin and increased IL-9, VEGF-α, and IP-10 associated with Aβ depositions in the gut. Aβ depositions was co-localized with blood vessels before central pathology in Tg 2576 AD mouse model.

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
