# Peer review of "Dysregulated Gut Homeostasis Observed Prior to the Accumulation of the Brain Amyloid-β in Tg2576 Mice"

_ijms, 2020, doi:10.3390/ijms21051711_

Round 1
Reviewer 1 Report
Dear Authors,
Congratulations for your work it is extremely interesting. For improving your manuscript please make some corrections:
1.correct your spelling and English typing
2. also I found more than 20% plagiarism rate on your manuscript but I think you can improve this issue.I attach the report.

Author Response
Thank you for the comments and suggestions.
Q1. correct your spelling and English typing
Ans: we went through and we could not find anything obvious
Q2. also I found more than 20% plagiarism rate on your manuscript but I think you can improve this issue.I attach the report.
The matched character strings are predominantly in the methods section, spelled out abbreviations, or authors’ emails. Nonetheless, we attempted to minimize the similarity by paraphrasing some of the highlighted sections.
Reviewer 2 Report
This paper describes a relatively new relationship between the intestinal barrier and the Aβ deposits related to Alzheimer disease (AD). In their animal model, many pathological signs appear before neuronal deposits are getting obvious. The question is very interesting, as the possibility emerges of early intervention especially if we look at the assumed role of Vitamin B12.
The paper is well written, the questions are answered clearly, and the methods are up-to-date.
I have only a few questions:
Is it known whether the AD diagnosed humans expressed the mutant form of APP?
As in the used animal model the double mutant form of APP is overexpressed under the control of a promoter originated from a prion protein, it is declared, that the protein is mainly neuron specific. Even if we accept that the protein appears in the blood, and will reach eventually the enterocytes, it is hard to imagine that the aggregates appear in these cells before they show up in the neurons. What could be the explanation for this? Moreover, how can a neuron specific protein initiate all the changes in the gut, which I do not question are real?
Author Response
Authors thank the reviewers for their excellent points.
Reviewer comments: The paper is well written, the questions are answered clearly, and the methods are up-to-date. I have only a few questions:
Q1: Is it known whether the AD diagnosed humans expressed the mutant form of APP?
Answer: Human AD samples were from sporadic AD patients. We clarified this in the results section.
Q2: As in the used animal model the double mutant form of APP is overexpressed under the control of a promoter originated from a prion protein, it is declared, that the protein is mainly neuron specific. Even if we accept that the protein appears in the blood, and will reach eventually the enterocytes, it is hard to imagine that the aggregates appear in these cells before they show up in the neurons. What could be the explanation for this? Moreover, how can a neuron specific protein initiate all the changes in the gut, which I do not question are real?
Answer: We added a short paragraph to the discussion to address this valid questions. In short, we speculate that luminal bacteria contribute to the formation of amyloid fibrils (“from biofilm”) which upon translocation contributes to systemic cascades of amyloidapathy.
Reviewer 3 Report
<Major issues>
Authors find novel findings Aβ deposition in ileum prior to cerebral Aβ deposition using model mouse and human patients with AD. These finding will be useful in clinical and basic research. In this study, authors attempt to elucidate the mechanisms focused on gut dysfunction in figure 4 and 5. But these data include critical problems and discrepancies.
This study has the robust data as following, 1) Aβ deposition in ileum prior to cerebral Aβ deposition in Tg2576 mice. 2) Tg2576 mice show the difference gut microbiota composition and gut dysfunction. 3) Young Tg2576 mice show the lacking of cubilin gene expression in ileum. But these three facts are seen independent in this manuscript. Does the different gut microbiota composition contribute the Aβ deposition in ileum, peripheral inflammation and cubilin gene expression? I cannot understand the importance of gut microbiota composition.
Although authors claimed that peripheral inflammatory events occur prior to development of cerebral Aβ deposition, plasma MCP-1 level was not increased in young age in figure 4. This is the serious discrepancy in this manuscript. Evaluation of other major pro-inflammatory cytokines including IL-1β and TNF-α may support authors’ hypothesis. In figure 5, decreasing cubilin gene expression in the young Tg2576 was not affect plasma VB12 levels, which indicated this expression change of cubilin is not functional at least in plasma VB12 levels. In addition, authors do not mention the mechanism in intestinal cubilin gene expression change in young Tg2576 mice. Hence, all data are seen independent. Finally, figure 8 should be removed because it is just authors’ speculation.
<Minor issues>
In statistical analyses, because this study has two factor including “genotype” and “age”, two-way ANOVA followed by post hoc test is proper. In addition, number of trials should be prepared at least five in all experiments.
Author Response
Reviewer: Authors find novel findings Aβ deposition in ileum prior to cerebral Aβ deposition using model mouse and human patients with AD. These finding will be useful in clinical and basic research. In this study, authors attempt to elucidate the mechanisms focused on gut dysfunction in figure 4 and 5. But these data include critical problems and discrepancies.
Reviewer Q: This study has the robust data as following, 1) Aβ deposition in ileum prior to cerebral Aβ deposition in Tg2576 mice. 2) Tg2576 mice show the difference gut microbiota composition and gut dysfunction. 3) Young Tg2576 mice show the lacking of cubilin gene expression in ileum. But these three facts are seen independent in this manuscript. Does the different gut microbiota composition contribute the Aβ deposition in ileum, peripheral inflammation and cubilin gene expression? I cannot understand the importance of gut microbiota composition.
Answer: We thank the reviewer for these critical points. We did see reduction in bacterial genus as early as pre-symptomatic timepoints of 6 months Tg2576 animals. Especially ruminiclostridium which is a SCFA producer is reduced compared to age-matched WT littermates. Interestingly we found drastic change in microbiota composition when we observed significantly high amyloid beta pathology in the brain. This data shows that some unknown immune factors coming from the host due to amyloid progression is responsible. However, we are yet to elucidate the factor. From our data set what we found is that cubilin transporter is lost in association with gut dysfunction. This lost in cubilin was associated with reduction in B12 availability at the symptomatic timepoint. Also, we are working on characterizing the immune responses in these animals that will give more mechanistic understanding of our data explained in the manuscript. However this is an observational study.
Reviewer Q: Although authors claimed that peripheral inflammatory events occur prior to development of cerebral Aβ deposition, plasma MCP-1 level was not increased in young age in figure 4. This is the serious discrepancy in this manuscript. Evaluation of other major pro-inflammatory cytokines including IL-1β and TNF-α may support authors’ hypothesis. In figure 5, decreasing cubilin gene expression in the young Tg2576 was not affect plasma VB12 levels, which indicated this expression change of cubilin is not functional at least in plasma VB12 levels. In addition, authors do not mention the mechanism in intestinal cubilin gene expression change in young Tg2576 mice. Hence, all data are seen independent. Finally, figure 8 should be removed because it is just authors’ speculation.
Answer: We thank the reviewer for pointing out these important points. We did look at 17 different cytokines and we only saw drastic changes in these 4 cytokines reported. Most of them were under the detection limit. MCP1 is usually known to be expressed high in AD conditions. In line with the previous findings our data also showed increased MCP1 symptomatic AD mice. MCP-1 usually was associated with amyloid aggregation, and in our study young mice did not have amyloid pathogenesis in the brain. We did see increased IP-10, IL-9 pro-inflammatory cytokines drastically increased in Tg pre-symptomatic mice (please see our discussion). We believe more information can be obtained by looking at the brain samples which we will follow up in our new ongoing studies. B12 is usually stored in the liver and it is very hard to detect the active B12. We believe that the deficiency of B12 can be only seen mush after the loss of transporter expression. we did add few points on cubilin mechanisms to better explain the gut dysfunction and cubilin expression. Figure 8 is made with the help of data obtained from the experiments carried-out. we did see these changes as described in the manuscript, mainly "we did see early gut dysfunction preceding amyloid beta deposition in the brain".
Reviewer: In statistical analyses, because this study has two factor including “genotype” and “age”, two-way ANOVA followed by post hoc test is proper. In addition, number of trials should be prepared at least five in all experiments.
Author: We agree with the reviewers that using two-way ANOVA is better for comparing different age group and genotype, however, in our entire study our main conclusive findings where we have only compared young Tg with young WT and Age Tg with Ag WT. That is the main reason for choosing one-way ANOVA with its multiple comparison correction. According to authors suggestion, we changed it to Two-way ANOVA with Tukey's or Sidak's multiple comparison test.
Round 2
Reviewer 3 Report
In revised manuscript, added reference (No.84) is very important and it can link authors’ novel finding including elevation of peripheral inflammation and cubilin expression change.
Although I still concern the different time point between reduction of cubilin expression and plasma VB12 levels, I believe that several problems will be solved in your future research. In the problems about pro-inflammatory, I can understand the undetectable level in other cytokines. Furthermore, statistical analyses were properly modified. This research includes important finding in clinical and basic research. I hope your development relating this research using these finding in near future.